# The Liquid Biopsy in the Management of Colorectal Cancer: An Overview

**DOI:** 10.3390/biomedicines8090308

**Published:** 2020-08-26

**Authors:** Marco Vacante, Roberto Ciuni, Francesco Basile, Antonio Biondi

**Affiliations:** Department of General Surgery and Medical-Surgical Specialties, University of Catania, Via S. Sofia 78, 95123 Catania, Italy; ciuni.r@gmail.com (R.C.); fbasile@unict.it (F.B.); abiondi@unict.it (A.B.)

**Keywords:** liquid biopsy, colorectal cancer, biomarkers, circulating tumor cells, circulating tumor DNA

## Abstract

Currently, there is a crucial need for novel diagnostic and prognostic biomarkers with high specificity and sensitivity in patients with colorectal cancer. A “liquid biopsy” is characterized by the isolation of cancer-derived components, such as circulating tumor cells, circulating tumor DNA, microRNAs, long non-coding RNAs, and proteins, from peripheral blood or other body fluids and their genomic or proteomic assessment. The liquid biopsy is a minimally invasive and repeatable technique that could play a significant role in screening and diagnosis, and predict relapse and metastasis, as well as monitoring minimal residual disease and chemotherapy resistance in colorectal cancer patients. However, there are still some practical issues that need to be addressed before liquid biopsy can be widely used in clinical practice. Potential challenges may include low amounts of circulating tumor cells and circulating tumor DNA in samples, lack of pre-analytical and analytical consensus, clinical validation, and regulatory endorsement. The aim of this review was to summarize the current knowledge of the role of liquid biopsy in the management of colorectal cancer.

## 1. Introduction

Colorectal cancer (CRC) is one of the most common solid cancers in developed countries, with approximately 1.8 million incident cases and 900,000 deaths every year worldwide [1,2]. The burden of CRC is growing in the majority of low- and middle-income countries, probably due to environmental risk factors, such as changes in diet and life-style (i.e., obesity, smoking, alcohol consumption, and suboptimal dietary habits) [3], aging, and urbanization [4,5]. According to the American Cancer Society (ACS), the 5-year survival rate ranges from 90% if CRC is diagnosed at a localized stage to 14% in patients presenting with metastatic disease [6]. Treatment decisions for CRC should take into account the stage of the disease, the general condition, and performance status of the patient, and the molecular characteristics of the tumor [7,8]. The diagnosis of CRC is frequently made using colonoscopy, and confirmed by histological examination of the tumor tissue biopsy. The TNM staging of CRC is based on the depth of invasion of the primary tumor, regional lymph node involvement, and distant metastases, which may contribute to the choice of the most appropriate therapeutic approach, including adjuvant chemotherapy [9]. Surgical resection with lymph node dissection represents the base of curative treatment for localized colon cancer. Patients with stage III colon cancer are treated with adjuvant therapy using the FOLFOX (leucovorin, 5-fluorouracil and oxaliplatin) regimen; however more data are needed to confirm the efficacy of such treatment for rectal cancer patients. Combination of doublet or triplet chemotherapy (i.e., 5-fluorouracil/leucovorin, capecitabine, oxaliplatin, irinotecan) and a targeted agent (i.e., cetuximab, bevacizumab, panitumumab) are routinely used for the treatment of metastatic CRC [10,11]. Histopathological tumor tissue analysis cannot be considered to be a reliable source of clinically helpful prognostic or predictive information for CRC at the individual patient’s level; thus, research is constantly moving towards the identification of more accurate and personalized biomarkers [12]. Indeed, there is a critical need for new diagnostic and prognostic biomarkers with high specificity and sensitivity in patients with CRC [13,14]. In this context, liquid biopsy could represent the new era for biomarkers detection: the term “liquid biopsy” refers to the isolation of cancer-derived components, such as circulating tumor cells (CTC), circulating tumor DNA (ctDNA), microRNAs (miRNAs), long non-coding RNAs (lncRNAs) and proteins, from peripheral blood or other body fluids (i.e., ascites, urine, pleural effusion, and cerebrospinal fluid), and their genomic or proteomic assessment [15,16]. Furthermore, exosomes (EXOs) which are membrane-bound extracellular vesicles containing proteins and nucleic acids released in the bloodstream by cancer cells, could represent potential biomarkers [17,18]. The aim of this review was to summarize the current knowledge of the role of liquid biopsy in the management of CRC.

## 2. Clinical Utility of Liquid Biopsies in Patients with Colorectal Cancer

Assessment of peripheral blood components, such as CTCs, ctDNA, miRNAs, and lncRNAs could improve CRC screening and diagnosis, and predict relapse and metastasis [19,20,21,22]. Blood-based liquid biopsies could also be effective in monitoring minimal residual disease (MRD) and drug resistance in CRC patients receiving chemotherapy [23,24] (Table 1).

### 2.1. Screening and Early Diagnosis

Global CRC screening guidelines recommend colonoscopy (every ten years), or flexible sigmoidoscopy (every five years) or fecal occult blood test (FOBT; every one or two years) for average-risk subjects aged 50–75 [62]. Blood-based detection tests represent an appealing alternative to these methods, as they are non-invasive and low-risk tests that can be easily performed during a routine medical check-up.

#### 2.1.1. Circulating Tumor Cells (CTC) and Circulating Endothelial Cell Clusters (ECC)

CTC detection is uncommon and rather difficult in early-stage CRC; thus, the utility of CTCs for CRC screening or early detection seems to be very poor [63]. However, a study by Tsai et al., carried out on 620 subjects (438 with adenoma, polyps, or stage I–IV CRC and 182 healthy controls) reported an overall accuracy of 88% for all tumor stages, including precancerous lesions, using a new CTC assay [25]. Tumor-derived circulating endothelial cell clusters (ECC) may represent a promising type of cell-based liquid biopsy for early detection of CRC. These circulating benign cell clusters are released directly from the tumor vasculature and their isolation and enumeration discriminated healthy subjects from treatment-naïve as well as pathological early-stage (≤IIA) CRC patients with high accuracy [64].

#### 2.1.2. Circulating Tumor DNA (ctDNA)

A recent meta-analysis concluded that the diagnostic accuracy of ctDNA has insufficient sensitivity but satisfactory specificity for diagnosis of CRC [23]. Nonetheless, there is growing evidence that ctDNA detection could be used along with the traditional screening methods to improve diagnosis of early-stage CRC [63,65,66]. In particular, a study by Flamini et al. showed that ctDNA, particularly when combined with carcinoembryonic antigen (CEA), may represent a useful tool for early detection of CRC (area under the ROC curve 0.92, with 84% sensitivity and 88% specificity) [37]. Combined assessment of ALU115, DNA integrity index (ALU247/115) and CEA could increase the diagnostic efficiency for CRC. Of note, serum DNA integrity index was superior to the absolute DNA concentration in diagnostic accuracy of CRC [38]. ctDNA methylation showed higher sensitivity compared to traditional serum tumor markers in early-stage CRC and could represent a potential diagnostic biomarker. Sun et al. showed that circulating, cell-free, methylated Septin 9 (mSEPT9) DNA had higher specificity than FOBT for the screening of CRC in 650 subjects (73% of CRC patients were mSEPT9-positive at 94.5% specificity, and 17.1% of patients with intestinal polyps and adenoma were mSEPT9-positive at 94.5% specificity) [39]. Furthermore, a recent prospective cohort study carried out on a high-risk population of 1493 subjects, demonstrated that a single ctDNA methylation marker, cg10673833, had high sensitivity (89.7%) and specificity (86.8%) for detection of precancerous lesions and CRC [67]. A meta-analysis by Nian et al. pointed out the efficacy of Epipro Colon 2.0 with 2/3 algorithm (Epigenomics), a test used to screen the methylation status of the SEPT9 promoter in ctDNA, for CRC detection. Positive ratio of mSEPT9 was higher in advanced CRC stages (45% in I, 70% in II, 76% in III, 79% in IV) and low differentiation tissue (31% in high, 73% in moderate, 90% in low). However, according to previous research, mSEPT9 did not seem to identify effectively precancerous lesions [68]. Other potential blood tests include a multi-analyte test (CancerSEEK) that could detect eight common solid tumor types, including CRC, through assessment of the levels of circulating proteins and mutations in ctDNA. The median test sensitivity was 73% for stage II, 78% for stage III and 43% for stage I tumors, with a specificity greater than 99% [69].

#### 2.1.3. Serum, Fecal, and Salivary MicroRNAs (miRNAs)

Alterations in miRNAs have been reported in blood or fecal samples from CRC patients, or even in subjects with precancerous advanced adenomas [40]. miRNAs can be observed in the circulation alone or combined with some proteins; also, they can be released directly into extracellular fluids and carried by microvesicles, mostly exosomes [70,71]. miR-129 is highly expressed in CRC plasma, while miR-24-2 levels are low in CRC serum, thus representing potential positive or negative biomarkers in the diagnosis of CRC patients [41,42]. A study showed that serum expression levels of five miRNAs (miR-31, miR-141, miR-224-3p, miR-576-5p and miR-4669) were significantly different between patients with colon cancer and healthy controls, suggesting their potential use as a miRNA panel for diagnosis of CRC [43]. miRNAs detection could be used to distinguish metastatic and non-mCRC patients. Indeed, high serum levels of miR-200c in CRC patients could potentially represent a predictive biomarker for local and distant metastasis [44]. A study demonstrated that exosomal miR-320d could significantly discriminate metastatic from non-mCRC patients with an AUC of 0.633 (95% CI: 0.526–0.740), the sensitivity of 62.0% and the specificity of 64.7%. The combination of miR-320d and CEA had an AUC of 0.804, with the sensitivity of 63.3% and the specificity of 91.3% [45]. Numerous miRNAs (i.e., miR-29a, miR-223, miR-224, miR-106a, and miR-135b) found in feces could represent useful biomarkers for screening and diagnosis of CRC [72]. Fecal miR-106a test combined with routine immunochemical FOBT have been reported to be effective in discriminating CRC patients from those with negative iFOBT results and could improve the sensitivity to identify CRC [46]. A study demonstrated that salivary miR-21 is significantly up-regulated in CRC patients with a very high sensitivity and specificity of 97 and 91% respectively, and could be an accurate biomarker for CRC screening [47]. More studies are needed to confirm if salivary miRNAs could represent reliable biomarker candidates for CRC detection [73].

### 2.2. Prognosis, Progression, and Response to Treatment

#### 2.2.1. Circulating Tumor Cells (CTC)

Several studies demonstrated that CTC could potentially play an important role in monitoring treatment outcomes and for detection of resistance against chemotherapy in CRC patients [26,27,28]. A prospective study by Bork et al. carried out on 287 patients with potentially curable CRC (including 239 patients with stage I–III) showed that preoperative CTC identification represented a strong and independent prognostic marker in non-mCRC [26]. In a cohort of 37 high-risk stages II–III CRC patients, Gazzaniga et al. pointed out that CTCs detection could facilitate the selection of high-risk stage II CRC patient candidates for adjuvant chemotherapy [27]. A study by Tsai et al. showed that rising counts of CTC in peripheral blood was associated with tumor progression and poor prognosis in CRC patients: CTC counts in 2 mL of peripheral blood increased from 0, 1, 5, to 36 in healthy (*n* = 27), benign (*n* = 21), non-metastatic (*n* = 95), and mCRC (*n* = 15) patients, respectively. After 2-year follow-up, non-mCRC patients who had ≥5 CTCs showed an 8-fold increased risk to develop metastasis within one year after curable surgery than those who had <5 CTC [28]. Furthermore, CTC could be used as tool for assessment of chemotherapy resistance [29,74]. High-toxicity multidrug regimens used against advanced CRC, often require the use of biomarkers to select the patients who will receive the most benefit. Stratification by CTC count was effective in detecting patients with previously untreated KRAS wild-type advanced CRC who could benefit the most from an intensive 4-drug protocol (oxaliplatin, irinotecan, and tegafur-uracil with leucovorin and cetuximab), avoiding high-toxicity treatment in low CTC groups [30]. A meta-analysis of 13 studies showed significant differences between CTC-low and CTC-high levels in CRC patients treated with chemotherapy with regard to disease control [Relative Risk (RR) = 1.354, 95% CI 1.002–1.830, *p* = 0.048], progression-free survival [PFS; Hazard Ratio (HR) = 2.500, 95% CI 1.746–3.580, *p* < 0.001] and overall survival (OS; HR = 2.856, 95% CI 1.959–4.164, *p* < 0.001). These results confirmed the prognostic and predictive role of CTCs for the response to chemotherapy in CRC patients [75].

#### 2.2.2. Circulating Tumor DNA (ctDNA)

The proportion of CRC patients in whom ctDNA can be identified depends on the tumor volume and ranges from 50% to 90% in those with non-metastatic or metastatic disease, respectively [76]. There is evidence that after CRC curative resection, the detection rate of ctDNA could range from 8–15% in stage II to 50% in stage IV [31,32,77]. Serum DNA concentrations and integrity index may play an important role not only in early complementary diagnosis but also in monitoring of progression and prognosis of CRC. A study showed that the median absolute serum ALU115 and ALU247/115 levels in patients with primary CRC were significantly higher than those in subjects with polyps or normal controls (*p* < 0.0001), in recurrent or metastatic CRC were significantly higher compared to primary CRC (*p* = 0.0021, *p* = 0.0018) or operated CRC (*p* < 0.0001, respectively) and during follow-up, ALU115 and ALU247/115 levels increased before surgery and reduced significantly after surgery [38]. A prospective study conducted on 53 metastatic colorectal cancer (mCRC) patients receiving standard first-line chemotherapy, showed that ctDNA is detectable in a high proportion of treatment-naïve mCRC patients, and early alterations in ctDNA during first-line chemotherapy could predict the later radiologic response. Significant decrease in ctDNA (median 5.7-fold; *p* < 0.001) levels were detected before cycle 2, which correlated with computerized tomography (CT) responses at 8–10 weeks [Odds Ratio (OR) = 5.25 with a 10-fold ctDNA reduction; *p* = 0.016]. Major decrease (≥10-fold) versus minor decrease in ctDNA precycle 2 was correlated with a trend for raised PFS (median 14.7 vs. 8.1 months; HR = 1.87; *p* = 0.266) [33]. Another prospective cohort study of 230 patients showed that detection of ctDNA after resection of stage II colon cancer may detect patients at very high risk of recurrence, thus giving direct evidence of residual disease and helpful information on adjuvant treatment choices. ctDNA was detected after surgery in 7.9% of patients who did not receive any adjuvant chemotherapy, and among these, 79% had recurred at a median follow-up of 27 months; recurrence was observed in 9.8% of 164 patients with negative ctDNA [HR = 18; 95% confidence interval (CI), 7.9 to 40; *p* < 0.001]. In patients who completed chemotherapy, the presence of ctDNA was correlated with a lower recurrence-free survival (HR = 11; 95% CI, 1.8 to 68; *p* = 0.001) [31]. ctDNA could detect the presence of residual metastatic cancer cells not evident on CT also in stage III CRC patients. Indeed, serial assessment of ctDNA could characterize subsets of patients benefiting or not benefiting from chemotherapy and represent a marker of adjuvant treatment efficacy [34,78]. It is well known that anti-epidermal growth factor receptor (EGFR) treatment is unsuccessful in the case of RAS mutations [79]. ctDNA detection could represent an alternative tool for selection of anti-EGFR treatment due to its agreement with mutational status of RAS in CRC tissue. A prospective-retrospective cohort study carried out on 146 mCRC patients, showed that plasma RAS assessment had high overall concordance and identified a mCRC population responsive to EGFR therapy with the same predictive level as standard of care PCR techniques tissue testing [35]. A prospective phase II clinical trial of cetuximab in RAS wild-type patients with CRC, combined sequential profiling of ctDNA and matched tissue biopsies with imaging and mathematical modeling of tumor progression, and showed that liquid biopsies were able to detect spatial and temporal heterogeneity of the resistance to anti-EGFR monoclonal antibodies [80]. In a phase II trial, the levels of RAS mutated ctDNA were assessed in mCRC patients treated with the oral multi-kinase inhibitor regorafenib. The reduction of RAS mutations in plasma within 8 weeks of therapy was associated with improved PFS and OS. Combination of dynamic contrast-enhanced magnetic resonance imaging (DCE-MRI) and ctDNA predicted duration of anti-angiogenic response and could improve management of patient treated with regorafenib [36]. A recent study by Siravegna et al. showed that plasma HER-2 (ERBB2) copy number analysis based on ctDNA could predict beneficial effects from HER-2-targeted therapy with high accuracy (97%) in 28 out of 29 patients [81].

#### 2.2.3. MicroRNAs (miRNAs)

Studies reported an association between high expression levels of specific miRNAs (including miR-21, miR-1290, miR-193a, miR-17-5p, miR-92a-3p, miR-203, miR-1229, and miR-17/92 cluster) and poor prognosis of CRC patients due to metastatic disease, post-treatment relapse, and poor OS [48,49,82,83,84,85,86]. On the other hand, low levels of serum exosomal miR-4772-3p and miR-6869-5p were associated with high risk of tumor recurrence in stage II and III and poor 3-year survival in CRC patients, respectively [50,51]. Furthermore, significantly higher expression of miR-6803-5p in CRC patients was associated with later TNM stage, lymph node or liver metastasis, and poor disease-free survival (DFS), thus representing a potential diagnostic and prognostic biomarker [51]. There is evidence that serum exosomal miR-21 could be a useful biomarker for the prediction of recurrence and poor prognosis at TNM stages II, III or IV in CRC patients [49]. Also, higher expression levels of serum exosomal miR-17-5p and miR-92a-3p predicted pathologic grades and stages of CRC [48]. A study reported that the exosomal miR-27a and miR-130a panel in plasma correlated with tumor grade and stage of CRC and could be effective for predicting poor OS (HR = 2.74; 95% CI, 1.25–6.01; *p* = 0.012; and HR = 2.36; 95% CI, 1.07–5.23; *p* = 0.034, respectively). Furthermore, both miRNAs could be used for detection of CRC: miR-27a showed a sensitivity of 82% and a specificity of 91%, while miR-130a showed a sensitivity of 70% and a specificity of 100% [52]. Serum exosomal miR-548c and miR-6803 could be important predictive biomarkers of DFS and OS in CRC patients. Indeed, studies showed that elevated levels of miR-6803 and decreased levels of miR-548c represented poor prognostic markers, particularly in later stages of CRC and in the presence of liver metastasis [51,53]. Specific miRNAs may be used for monitoring resistance or tolerance to chemotherapy and for selection of clinical therapeutic approach. A panel of serum exosomal miRNAs including miR-1246, miR-21-5p, miR-1229-5p, and miR-96-5p could significantly discriminate chemotherapy-resistant subjects to 5-FU and oxaliplatin from advanced CRC patients (AUC = 0.804; *p* < 0.05). Targeting these miRNAs could enhance chemosensitivity to oxaliplatin and 5-FU, thus representing a promising approach for CRC treatment [54]. A study by Yagi et al. suggested that increased plasma exosomal miR-125b levels could detect resistance to modified FOLFOX6-based first-line chemotherapy in patients with advanced or recurrent CRC. Furthermore, PFS was significantly inferior in patients with high miR-125b levels before chemotherapy than in those with low levels, thus confirming the utility of miR-125b as a predictive biomarker in advanced or recurrent CRC [55].

#### 2.2.4. Long Non-Coding RNAs (lncRNAs)

lncRNAs interact with DNA, mRNA, proteins, and miRNAs, playing a role in multiple biological processes, such as epigenetic or gene expression regulation, and chromatin remodeling [87,88]. Several studies showed that lncRNAs were abnormally expressed in many cancers, including CRC, and therefore could have potential application in diagnosis, prognosis and potential treatment [89,90,91,92]. Indeed, lncRNAs could regulate drug function and chemoresistance through different mechanisms in many tumors, including CRC [56,57]. More than 70 CRC-related lncRNAs have been identified so far, including HOTAIR, MEG3, CRNDE, UCA1, CCAT1, CCAT2, MALAT-1 and H19 [93]. Alterations in the expression of these lncRNAs could lead to chemotherapy and radiotherapy resistance. Sun et al. identified four hub lncRNAs (CRNDE, H19, UCA1, and HOTAIR) involved in the process of resistance to oxaliplatin or irinotecan in patients with advanced CRC. In particular, high expression of HOTAIR was associated with advanced and metastatic disease and poor prognosis [58]. Decreased serum MEG3 levels were correlated with poor response to chemotherapy and OS in CRC patients treated with oxaliplatin. MEG3 increased oxaliplatin-induced cell apoptosis in CRC; therefore, overexpression of MEG3 could represent a promising therapeutic strategy to defeat oxaliplatin resistance in CRC patients [57]. Tang et al. demonstrated that up-regulation of a lncRNA, GLCC1, under glucose-limited conditions in CRC cells, promoted cell survival and proliferation by stabilizing c-Myc and stimulating glycolysis. From a clinical point of view, GLCC1 was associated with carcinogenesis, tumor volume and poor prognosis in CRC patients [59,94]. Levels of serum exosomal CRNDE-h were higher in CRC patients compared to those with benign colorectal disease or healthy controls. CRNDE-h expression could be related to the presence of lymph node metastasis and was associated with a low OS in CRC. Furthermore, the prognostic value of CRNDE was better than CEA, with a sensitivity of 70% vs. 37% and a specificity of 94% vs. 89% [60]. Liang et al. reported that high exosomal RPPH1 levels were associated with advanced TNM stages, promotion of metastasis, and poor prognosis in CRC patients, whereas lower RPPH1 levels were observed after tumor resection. Plasma exosomal RPPH1 levels showed a better diagnostic value (AUC = 0.86) compared to CEA and CA19.9 [61]. A study by Barbagallo et al. demonstrated that UCA1 was down-regulated in serum of CRC patients compared to healthy subjects; UCA1 showed an AUC of 0.719 (95% CI, 0.533–0.863; *p* = 0.01) with 100% sensitivity and 43% specificity in discriminating between the cancer and control groups. These results suggested that the UCA1 regulatory axis could be a promising target to develop novel RNA-based therapies against CRC [95].

## 3. Current Issues and Limitations of Liquid Biopsy

Despite all the potential advantages of liquid biopsy in the management of CRC, there are still some practical issues that need to be addressed before it can be widely used in clinical practice [96]. Potential challenges may include low amounts of CTCs and ctDNA in samples, lack of pre-analytical and analytical consensus, clinical validation, regulatory endorsement and cost effectiveness [97,98]. Currently, the use of CTCs in routine diagnostics is limited, mainly due to methodological constraints, such as the lack of an established assessment practice, beyond enumeration [99,100]. The epithelial cell adhesion molecule (EpCAM)-dependent technique was approved by the U.S. Food and Drug Administration (FDA) in 2004, and represents the “gold standard” for CTC isolation in different cancers, including CRC [101]. However, only CTCs that maintain epithelial features can be detected by EpCAM, excluding CTCs with mesenchymal characteristics [102]. On the other hand, ctDNA analysis has been better optimized for routine diagnostic use [103]. The concentration of ctDNA in the peripheral blood depends on the site, volume, and vascularity of the tumor, which can also be responsible for the large variations frequently observed in ctDNA levels [104]. Analysis of ctDNA can be performed by either quantitative assessment of ctDNA in a blood sample or by the identification of mutations. The introduction of next-generation sequencing (NGS)-based technologies reduced the error rate and enhanced sensitivity in ctDNA detection [105]. NGS technology enables the analysis of thousands of DNA sequences in parallel followed by either sequence alignment to a reference genome or de novo sequence assembly [104,106]. Deep sequencing represents the first approach to identify mutations at a low allele frequency (<0.2%) by sequencing the target regions with high coverage (>10,000×) [107]. Therefore, the sensitivity of deep sequencing for detecting mutations in ctDNA can achieve 100%, even if the specificity can be lower, around 80% [108,109]. Advantages of NGS included detection of genomic rearrangements, new mutations or alterations in genes, and the possible evaluation of response to treatment [110]. However, NGS-based approaches are rather expensive and time-consuming. Furthermore, data should be analyzed and interpreted by experts in bioinformatics [111]. Data storage and the difficulty in interpreting massive quantity of information obtained with NGS may represent a computational challenge to researchers. Also, the selection of proper validation methods to detect clinically significant mutations among a large number of samples can represent a challenging task [112]. Clinical validation of NGS data is carried out by assessing various parameters such as analytical sensitivity (the ability of the test to identify true sequence variants e.g., false negative rate), and analytical specificity (the probability of the test to not identify mutations where none are present (e.g., false positive rate) [113]. Limitations of NGS, principally with regard to the overall clinical sensitivity, could be overtaken implementing NGS with mutant allele enrichment or using digital PCR to improve reliability [96,114]. Mass-spectrometry and Real-Time PCR are other promising techniques for ctDNA assessment, which are rapid and cheap, require small quantities of input material, and have high sensitivity and specificity [77,115]. If possible, ctDNA should be analyzed in combination with CTCs and exosomal miRNAs, to obtain as much data as possible from a single blood sample [116]. However, different blood collection tubes, changes in storage temperatures and centrifugation may affect DNA or cells stability [117,118,119]. ctDNA degradation due to DNase activity could be avoided by isolating plasma within an hour after blood draw [120]. Reduction of cell lysis and stabilization of the total ctDNA pool can be obtained by means of specific blood collection tubes containing preservatives and additives [121]. Furthermore, accuracy and reproducibility of the liquid biopsy represent a main issue for analytic validity [122]. A study by Vivancos et al. showed that two liquid biopsy platforms, OncoBEAM™ RAS CRC and Idylla™ ctKRAS Mutation Test, had different sensitivity for identifying KRAS mutations in plasma samples from mCRC patients. The European Molecular Genetics Quality Network (EMQN) evaluated ctDNA detection approaches, and underlined that multiple pre-analytical and analytical variants may produce variable results; the EMQN pilot external quality assessment (EQA) scheme showed that the existing variability in multiple phases of ctDNA processing and analysis (e.g., due to specimen volume, ctDNA quantification technique, and choice of genotyping platform), resulted in an overall error rate of 6.09% [123]. These results highlighted the critical need for better standardization and validation of liquid biopsy assessment [124].

## 4. Future Perspectives and Conclusions

The use of CTCs, ctDNA, miRNAs and lncRNAs as potential biomarkers is an emerging area with a great potential for the management of CRC. Currently, the clinical utility of liquid biopsies in CRC limited, but it is expected to achieve a clear consensus in the near future. Indeed, the liquid biopsy is a minimally invasive, cheap, and repeatable technique that can facilitate CRC screening and early diagnosis, providing more information for the clinical staging of CRC patients. Furthermore, blood-based liquid biopsies are useful for monitoring disease progression and treatment efficacy, prognosis, and acquired resistance to chemotherapy in CRC. It is reasonable to think that in the future, it will be possible to choose the most appropriate therapy based on real-time genetic information through a liquid biopsy, in the way of personalized medicine. In this context, performing prospective clinical trials is essential for clinical utility and development of practice changing protocols. Nevertheless, the transfer of liquid biopsies from bench to bedside necessitates larger-scale and multicenter trials to confirm its advantages. Also, optimization of pre-analytical and analytical processing is fundamental for clinical validity, and standardization of laboratory methods is firmly required to guarantee elevated reproducibility of the results. The lack of clinical applicability is currently due to large quantity of liquid biopsy assays. For example, many ctDNA assays are presently commercially available, but each assay shows specific detection limit, sensitivity, and specificity. Therefore, the results obtained from different liquid biopsy platforms cannot be easily compared, and EQA studies are needed before application in routine diagnostics. Further studies should be conducted on the effectiveness of liquid biopsy biomarkers, such as ctDNA, in combination with other blood tests and radiological monitoring, in order to better identify and stratify CRC patients and to choose the appropriate treatment. In the future, advances in liquid biopsy methodologies and their increased sensitivity should facilitate detection of MRD and early CRC diagnosis even in asymptomatic subjects. Only a few trials have investigated a specific intervention based on the results of liquid biopsies (i.e., CTC or ctDNA status), so far. Many of these studies did not include a control group, and therefore the results could not lead to significant changes in clinical practice. Further prospective studies are needed to establish future clinical applications of liquid biopsies and delineate their impact in the management of CRC.

## Figures and Tables

**Table 1 biomedicines-08-00308-t001:** Potential clinical applications of liquid biopsy biomarkers in CRC.

Study (Year)	Biomarkers	Sample Size	Methods	Statistical Significance (*p* Value), Sensitivity/Specificity (%) and/or Hazard Ratio	Potential Clinical Applications
Tsai et al. (2018) [25]	CTC	*n* = 620 (*n* = 438 adenoma, polyps, or stage I–IV CRC, *n* = 182 healthy controls).	CellMax biomimetic platform (CMx)	All subjects: Sn 84.0/Sp 97.3 Precancerous lesions: Sn 76.6/Sp 97.3 CRC: Sn 86.9/Sp 97.3	Screening
Bork et al. (2015) [26]	CTC	Total *n* = 287 (*n* = 239 stage I–III CRC)	CellSearch	OS: HR 5.5 (95% CI 2.3–13.6, *p* < 0.001) PFS: HR 12.7 (95% CI 5.2–31.1, *p* < 0.001)	Prognostic in non-mCRC
Gazzaniga et al. (2013) [27]	CTC	*n* = 37 high-risk stage II or III CRC	CellSearch	The presence of CTC was detected in 8 of 37 patients (22%) 87.5% of CTC-positive patients had N1–2 disease and stage III CRC	Selection of high-risk stage II CRC patient candidates for adjuvant chemotherapy
Tsai et al. (2016) [28]	CTC	*n* = 158 (*n* = 27 healthy, *n* = 21 benign, *n* = 95 non-mCRC, *n* = 15 m-CRC)	CellMax biomimetic platform (CMx)	CRC: Sn 63.0/Sp 82.0 All colorectal neoplasms, including adenomatous polyps, dysplastic polyps, and CRC: Sn 61.0/Sp 94.0	Prognostic in non-mCRC at high risk of early recurrence
Musella et al. (2015) [29]	CTC	*n* = 38 advanced RAS-BRAF-wild-type CRC receiving third-line therapy with cetuximab-irinotecan or panitumumab.	AdnaTest ColonCancerSelect	OS: HR 8.06 (95% CI, 2.54–25.59, *p* < 0.001) PFS: HR 6.10 (95% CI, 2.49–14.96, *p* < 0.001)	Prognostic and predictive in CRC patients treated with anti-EGFR monoclonal antibodies
Krebs et al. (2014) [30]	CTC	*n* = 48 (CTC enumeration performed only in 42 patients)	CellSearch	ORR: 71% Median OS for high and low CTC count: 18.7 and 22.3 months (log-rank test, *p* < 0.038)	Prognostic in CRC patients treated with irinotecan, oxaliplatin, and tegafur-uracil with leucovorin and cetuximab
Tie et al. (2016) [31]	ctDNA	*n* = 230 resected stage II colon cancer	Safe-SeqS	Postoperative recurrence at 36 months: Sn 48.0/Sp 100.0	Monitoring of MRD and identification of CRC patients at very high risk of recurrence
Sun et al. (2018) [32]	ctDNA	*n* = 11 CRC treated surgically	NGS	*n* = 7: decreased mutation rates in postoperative vs. preoperative period *n* = 4: no mutations *n* = 1 patient with metastatic rectal cancer: the rate of TP53 mutation increased from 8.95 (preoperative) to 71.4% (postoperative)	Prognostic and Predictive
Tie et al. (2015) [33]	ctDNA	*n* = 53 mCRC patients receiving standard first-line chemotherapy	Safe-SeqS	10-fold change ctDNA threshold: Sn 75.0/Sp 64.0	Predictive during first-line chemotherapy
Tie et al. (2018) [34]	ctDNA	*n* = 95 stage III colon cancer receiving adjuvant chemotherapy	Safe-SeqS	Inferior RFS: in case of positive ctDNA post-surgery (HR 3.52, *p* = 0.004). Superior RFS: when ctDNA became undetectable after chemotherapy (HR 5.11, *p* = 0.02). Inferior RFS: when ctDNA status changed from negative to positive after chemotherapy (HR 5.30, *p* = 0.006). Inferior RFS: positive ctDNA after adjuvant chemotherapy completion (HR 7.14, *p* < 0.001)	Prognostic and therapy monitoring in stage III colon cancer
Grasselli et al. (2017) [35]	ctDNA	*n* = 146 mCRC patients	SoC PCR and Digital PCR (BEAMing)	ctDNA BEAMing RAS testing showed 89.7% agreement with SoC (Kappa index 0.80, 95% CI 0.71–0.90) BEAMing in tissue showed 90.9% agreement with SoC (Kappa index 0.83, 95% CI 0.74–0.92)	Predictive and anti-EGFR treatment selection
Khan et al. (2018) [36]	ctDNA	*n* = 27 RAS mutant mCRC	Digital-droplet PCR	PFS: HR 0.21 (95% CI 0.06–0.71, *p* = 0.01)	Predictive of duration of anti-angiogenic response to regorafenib
Flamini et al. (2006) [37]	ctDNA	*n* = 75 healthy subjects *n* = 75 CRC	qPCR	ctDNA alone: Sn 81.3/Sp 73.3 ctDNA + CEA: Sn 84.0/Sp 88.0	Diagnosis of early-stage CRC
Hao et al. (2014) [38]	ctDNA	*n* = 104 primary CRC, *n* = 85 operated CRC, *n* = 16 recurrent/mCRC, *n* = 63 intestinal polyps, *n* = 110 normal controls	ALU-qPCR	ALU115: Sn 69.23/Sp 99.09 ALU247/115: Sn 73.08/Sp 97.27	Early complementary diagnosis, monitoring of progression and prognosis of CRC
Sun et al. (2019) [39]	mSEPT9 DNA	*n* = 650	Epigenomics AG for Epi proColon 2.0	CRC: Sn 73.0/Sp 94.5 Polyps and adenoma: Sn 17.1/Sp 94.5	Screening and recurrence monitoring
Link et al. (2010) [40]	Fecal miRNAs	*n* = 8 healthy controls, *n* = 29 normal colonoscopies, colon adenomas, and CRCs	TaqMan qRT-PCR	Increased expression of miR-21 and miR-106a in CRC and adenomas vs. normal controls (*p* < 0.05)	Screening
Ya et al. (2017) [41]	Serum miR-129	*n* = 18 female patients with CRC	Real-time PCR	Contribution to carcinogenesis by targeting ERβ (*p* < 0.01)	Development of therapeutic agents
He et al. (2018) [42]	Serum miR-24-2	*n* = 68 healthy subjects, *n* = 228 CRC	Real-time qRT-PCR	Higher levels in CRC than healthy subjects (*p* < 0.05)	Negative biomarker in the diagnosis of the progression of CRC
Wang et al. (2017) [43]	Serum miR-31, miR-141, miR-224-3p, miR-576-5p, and miR-4669	*n* = 44 healthy subjects, *n* = 50 CRC. Double-blind validation using sera from 30 CRC, 30 colonic polyps, 30 healthy controls	Real-time PCR	AUC = 0.995 (microarrays) AUC = 0.964 (double-blind validation test)	Panel for diagnosis of CRC
Toiyama et al. (2014) [44]	Serum miR-200c	Total *n* = 446 colorectal specimens. First phase: *n* = 12 stage I and IV CRC. Second phase: *n* = 182 CRC, *n* = 24 controls. Third phase: *n* = 156 tumor tissues from 182 CRC and an independent set of 20 matched primary CRC and corresponding liver mts	Real-time qRT-PCR	Correlation with lymph node mts (*p* = 0.0026), distant mts (*p* = 0.0023), and prognosis *(p* = 0.0064) Predictor for lymph node mts (OR 4.81, 95% CI 1.98–11.7, *p* = 0.0005) and tumor recurrence (HR 4.51, 95% CI 1.56–13.01, *p* = 0.005) Prognostic (HR 2.67, 95% CI 1.28–5.67, *p* = 0.01)	Prognostic and predictive of metastasis
Tang et al. (2019) [45]	Exosomal miR-320d	*n* = 34 mCRC, *n* = 108 non-mCRC	qPCR	miR-320d: AUC = 0.633, *p* = 0.019 miR-320d + CEA: AUC = 0.804	Predictive of metastasis
Koga et al. (2013) [46]	Fecal miR-106a	*n* = 117 CRC, *n* = 107 healthy subjects	Real-time RT-PCR	FmiRT: Sn 34.2/Sp 97.2. iFOBT + FmiRT: Sn 70.9/Sp 96.3	Screening
Sazanov et al. (2017) [47]	Plasma and saliva miR-21	Plasma: total *n* = 65 CRC (*n* = 34 controls, *n* = 6 stage II, *n* = 16 stage III, *n* = 9 stage IV) Saliva: total *n* = 68 CRC (*n* = 34 controls, *n* = 6 stage II, *n* = 18 stage III, *n* = 10 stage IV)	Real-time qRT-PCR	Plasma: Sn 65/Sp 85 Saliva: Sn 97/Sp 91	Screening
Fu et al. (2018) [48]	Exosomal miR-17-5p and miR-92a-3p	*n* = 10 normal controls, *n* = 18 CRC, *n* = 11 mCRC	Real-time qPCR	miR-17-5p: AUC = 0.897 (95% CI 0.800–0.994) for CRC, and 0.841 (95% CI 0.720–0.962) for mts miR-92a-3p: AUC = 0.845 (95% CI 0.724–0.966) for CRC and 0.854 (95% CI 0.735–0.973) for mts miR-17-5p + miR-92a-3p: AUC = 0.910 (95% CI 0.820–1) for CRC and 0.841 (95% CI 0.718–0.964) for mts	Prognostic
Tsukamoto et al. (2017) [49]	Exosomal miR-21	Total *n* = 326 CRC (*n* = 51 stage I, *n* = 110 stage II, *n* = 98 stage III, *n* = 67 stage IV)	TaqMan miRNA assays	OS: HR 2.28 (95% CI 1.81–5.74, *p* < 0.01) DFS: HR 2.34 (95% CI 1.87–4.60, *p* < 0.01)	Prediction of recurrence and poor prognosis in CRC patients with TNM stage II, III, or IV
Liu et al. (2016) [50]	Exosomal miR-4772-3p	*n* = 84 stage II–III colon cancer	Real-time qRT-PCR	AUC = 0.72 (95% CI 0.59–0.85, *p* = 0.001)	Prognostic for tumor recurrence in stage II and III colon cancer patients
Yan et al. (2018) [51]	Exosomal miR-6803-5p	*n* = 168 CRC	qRT-PCR	OS: HR 2.93 (95% CI 1.35–6.37, *p* < 0.007) DFS: HR 3.26 (95% CI 1.56–6.81, *p* < 0.002) AUC = 0.7399	Diagnostic and prognostic
Liu et al. (2018) [52]	Exosomal miR-27a and miR-130a	Training phase: *n* = 40 healthy subjects *n* = 40 stage I CRC. Validation phase: *n* = 40 stage I, n = 20 stage II, *n* = 14 stage III, *n* = 6 stage IV CRC, *n* = 40 healthy subjects. External validation phase: 50 stage I CRC, 50 adenomas, 50 healthy subjects	qRT-PCR	miR-27a: AUC = 0.773 Sn 75/Sp 77.5 in the training phase, AUC = 0.82 Sn 80.0/Sp 77.5 in the validation phase, and AUC = 0.746 Sn 80.0/Sp 77.5 in the external validation phase miR-130a: AUC = 0.742 Sn 82.5/Sp 62.5 in the training phase, AUC = 0.787 Sn 70.0/Sp 80.0 in the validation phase, AUC = 0.697 Sn 70.0/Sp 80.0 in the external validation phase miR-27a + miR-130a: training phase AUC = 0.846 Sn 82.5/Sp 75, validation phase AUC = 0.898, Sn 80.0/Sp 90.0 and external validation phase AUC = 0.801 Sn 80.0/Sp 90.0	Diagnostic and prognostic
Peng et al. (2018) [53]	Exosomal miR-548c-5p	*n* = 108 CRC	Real-time qPCR	OS: HR 3.40 (95% CI 1.02–11.27, *p* = 0.046)	Diagnostic and prognostic
Jin et al. (2019) [54]	Exosomal miR-21-5p, miR-1246, miR-1229-5p, and miR-96-5p	Drug-resistant CRC cell lines	qRT-PCR	AUC = 0.804, *p* < 0.05	Predictive for chemoresistance in advanced CRC
Yagi et al. (2019) [55]	Exosomal miR-125b	*n* = 55 patients with advanced/recurrent CRC treated with mFOLFOX6	qRT-PCR	PFS: HR 0.71 (95% CI 0.36–0.94, *p* < 0.041)	Predictive and detection of chemotherapy resistance
Wang et al. (2018) [56]	lncRNA H19	*n* = 110 paired CRC tissues and para-tumor tissues	qRT-PCR	RFS: log-rank test *p* < 0.001 High H19: HR 2.383 (95% CI 1.157–4.909, *p* = 0.018)	Predictive of 5-FU resistance
Li et al. (2017) [57]	lncRNA MEG3	*n* = 316 CRC	qRT-PCR	AUC = 0.784, Sn 72.86/Sp 61.43 OS: HR 1.390 (95% CI 0.324–2.089, *p* = 0.007)	Prognostic and promotion of chemosensitivity
Sun et al. (2019) [58]	lncRNAs CRNDE, H19, UCA1, and HOTAIR	CRC cell lines (HCT116, HT29, and LoVo)	Gene Expression Profiling Interactive Analysis	HOTAIR OS: HR 1.9, *p* = 0.0066 DFS: HR 1.8, *p* = 0.012	Predictive of treatment sensitivity
Tang et al. (2019) [59]	lncRNA GLCC1	In vitro: Human colorectal cancer cell lines SW1116, SW480, Caco2, LoVo, HT29, RKO, DLD-1, and HCT116 In vivo: BALB/c nude mice	Real-time qPCR	Stabilization of c-Myc after knockdown of lncGLCC1 (*p* < 0.001)	Prognostic
Liu et al. (2016) [60]	Exosomal lncRNA CRNDE-h	*n* = 468	qRT-PCR	AUC = 0.892 Sn 70.3/Sp 94.4	Diagnostic and prognostic
Liang et al. (2019) [61]	Exosomal lncRNA RPPH1	*n* = 61 CRC	qRT-PCR	OS: HR 2.145 (95% CI 1.450–3.174, *p* < 0.001) DFS: HR 1.820 (95% CI 1.257–2.637, *p* = 0.001)	Prognostic, therapeutic, and diagnostic target

CTC: circulating tumor cells; UICC: Union for International Cancer Control; HR: hazard ratio; Sn: sensitivity; Sp: Specificity; mCRC: metastatic colorectal cancer; ctDNA: circulating tumor DNA; OS: overall survival; PFS: progression-free survival; ORR: objective response rate; EGFR: epidermal growth factor receptor; MRD: minimal residual disease; RFS: recurrence-free survival; CI: confidence interval; CEA: carcinoembryonic antigen; NGS: next-generation sequencing; SoC: Standard of care; qPCR: quantitative polymerase chain reaction; qRT-PCR: quantitative reverse transcription polymerase chain reaction; mSeptin9: methylated septin9; ERβ: estrogen receptor β; AUC: area under the ROC (Receiver Operating Characteristic) curve; OR: odds ratio; iFOBT: immunochemical fecal occult blood test; FmiRT: fecal microRNA test; mts: metastasis; DFS: disease-free survival.

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
