# Peer review of "The Liquid Biopsy in the Management of Colorectal Cancer: An Overview"

_biomedicines, 2020, doi:10.3390/biomedicines8090308_

Round 1

Reviewer 1 Report

The review manuscript titled “The Liquid Biopsy in the Management of Colorectal Cancer: An Overview” focuses on the current knowledge on the role of liquid biopsy in the management of colorectal cancer. The authors presented a comprehensive review detailing both advantages and limitations of liquid biopsy, and also discussed the potential of liquid biopsy for the management of CRC. This review would be of interest to researcher in cancer biology.

Best of luck.

Author Response

Reviewer 1

The review manuscript titled “The Liquid Biopsy in the Management of Colorectal Cancer: An Overview” focuses on the current knowledge on the role of liquid biopsy in the management of colorectal cancer. The authors presented a comprehensive review detailing both advantages and limitations of liquid biopsy, and also discussed the potential of liquid biopsy for the management of CRC. This review would be of interest to researcher in cancer biology.

Best of luck.

Reply: Many thanks for your peer review and your kind comments.

Reviewer 2 Report

In their review “The Liquid Biopsy in the Management of Colorectal Cancer: An Overview” the authors provide a compilation of studies investigating the utility of several types of liquid biopsies for the clinical management for colorectal carcinoma. The authors also briefly discuss the limitations and challenges of liquid biopsies.

The abstract is well written, concise and informative and will be of value for the community. Therefore I recommend publication in general, but I have two suggestions that I think would add a lot of value.

  1. The section about limitations and challenges of liquid biopsies is kept very brief. This section would greatly benefit from a bit more detail. For example, it is mentioned that NGS increases sensitivity but why and how is left open. Along these lines at least touching on the analytical challenges and how they are addressed especially for NGS would be beneficial.
  2. Add reported sensitivity/specificity in each study to the table and/or add a diagram (scatter-plot) positioning the discussed methods/studies according to their sensitivity/specificity in order to provide a more direct/visual overview.

Author Response

Reviewer 2

In their review “The Liquid Biopsy in the Management of Colorectal Cancer: An Overview” the authors provide a compilation of studies investigating the utility of several types of liquid biopsies for the clinical management for colorectal carcinoma. The authors also briefly discuss the limitations and challenges of liquid biopsies.

The abstract is well written, concise and informative and will be of value for the community. Therefore I recommend publication in general, but I have two suggestions that I think would add a lot of value.

The section about limitations and challenges of liquid biopsies is kept very brief. This section would greatly benefit from a bit more detail. For example, it is mentioned that NGS increases sensitivity but why and how is left open. Along these lines at least touching on the analytical challenges and how they are addressed especially for NGS would be beneficial.

Reply: We revised the section about limitations and challenges of liquid biopsies (“3. Current Issues and Limitations of Liquid Biopsy”) and added more discussion on how and why NGS may increase sensitivity (lines 295-301). As suggested, we also discussed the issues associated with analytical challenges, in particular for NGS, and their possible management (lines 304-313, lines 318-321, and lines 324-330).

Add reported sensitivity/specificity in each study to the table and/or add a diagram (scatter-plot) positioning the discussed methods/studies according to their sensitivity/specificity in order to provide a more direct/visual overview.

Reply: We modified table 1 and added a column reporting information on statistical significance, hazard ratio, and/or sensitivity/specificity values for each study (where available).

Reviewer 3 Report

This review presents updated information on the development of liquid biopsy techniques in relation to colorectal cancer. The information is presented in a concise and structured manner around several biomarkers that have shown potential utility in the diagnosis, prognosis, and therapeutic follow-up of patients with colorectal cancer.

The conclusions reached are well reasoned and show the points that must be stressed so that these techniques become routine in clinical practice.

I would like to make two suggestions to the authors that could improve the readability of a part of the manuscript and provide interesting information so that the reader can better interpret the information contained in the paper:

  • Section 2.1. Screening and Early Diagnosis (pages 4-5, lines 70 to 130) is very dense and hard to follow as it is written. It would be advisable to separate the lines dedicated to each biomarker considered (CTCs, ctDNA, miRNAs, lncRNAs) into separate subsections as is done in the subsequent section.
  • It would be interesting to include in Table 1 a column that reflects the sample size of each study contained in it. This information is cited in the text for some explicitly commented studies, but if it appeared in the table for the 36 referenced studies, it would provide the reader with a minimum criterion to assess the relative importance of each one.

Author Response

Reviewer 3

This review presents updated information on the development of liquid biopsy techniques in relation to colorectal cancer. The information is presented in a concise and structured manner around several biomarkers that have shown potential utility in the diagnosis, prognosis, and therapeutic follow-up of patients with colorectal cancer.

The conclusions reached are well reasoned and show the points that must be stressed so that these techniques become routine in clinical practice.

I would like to make two suggestions to the authors that could improve the readability of a part of the manuscript and provide interesting information so that the reader can better interpret the information contained in the paper:

  • Section 2.1. Screening and Early Diagnosis (pages 4-5, lines 70 to 130) is very dense and hard to follow as it is written. It would be advisable to separate the lines dedicated to each biomarker considered (CTCs, ctDNA, miRNAs, lncRNAs) into separate subsections as is done in the subsequent section.

Reply: As suggested, we divided the paragraph 2.1. into different subparagraphs for each biomarker discussed.

  • It would be interesting to include in Table 1 a column that reflects the sample size of each study contained in it. This information is cited in the text for some explicitly commented studies, but if it appeared in the table for the 36 referenced studies, it would provide the reader with a minimum criterion to assess the relative importance of each one.

Reply: We modified table 1 and added a column reporting data from the sample size and the subjects included in the subgroups (e.g. CRC, healthy controls) for each study.